# Online Learning for Reference-Based Super-Resolution

Byungjoo Chae [1,†], Jinsun Park [2,†] , Tae-Hyun Kim [3] and Donghyeon Cho [1,*]

1   Department of Electronics Engineering, Chungnam National University, 99 Daehak-ro, Yuseong-gu, Daejeon 34134, Korea; wpdhkd1642@gmail.com
2   School of Computer Science and Engineering, Pusan National University, 2 Busandaehak-ro 63beon-gil, Geumjeong-gu, Busan 46241, Korea; jspark@pusan.ac.kr
3   Department of Computer Science, Hanyang University, 222 Wangsimni-ro, Seongdong-gu, Seoul 04763, Korea; taehyunkim@hanyang.ac.kr
*   Correspondence: cdh12242@cnu.ac.kr; Tel.: +82-42-821-5667
†   These authors contributed equally to this work.

**Abstract:** Online learning is a method for exploiting input data to update deep networks in the test stage to derive potential performance improvement. Existing online learning methods for single-image super-resolution (SISR) utilize an input low-resolution (LR) image for the online adaptation of deep networks. Unlike SISR approaches, reference-based super-resolution (RefSR) algorithms benefit from an additional high-resolution (HR) reference image containing plenty of useful features for enhancing the input LR image. Therefore, we introduce a new online learning algorithm, using several reference images, which is applicable to not only RefSR but also SISR networks. Experimental results show that our online learning method is seamlessly applicable to many existing RefSR and SISR models, and that improves performance. We further present the robustness of our method to non-bicubic degradation kernels with in-depth analyses.

**Keywords:** reference-based SR; online learning; self-supervised learning

## 1. Introduction

Deep learning-based single-image super-resolution (SISR) algorithms [1–9] have shown remarkable progress in recent years. However, these algorithms still suffer from blurry output images because they are generally trained to minimize the mean squared error (MSE) or mean absolute error (MAE) between the network output and ground truth images. This problem has led to various efforts to generate high-frequency details with a generative adversarial network (GAN) and/or perceptual losses [10–12]. However, these methods often lead to reduced reconstruction performance with unexpected visual artifacts. The reason for this is that GAN-based deep networks often generate visually pleasing images, but fail to recover genuine information lost during the downsampling (degradation) process. In order to reconstruct the lost information, reference-based super-resolution (RefSR) methods have been proposed. RefSR algorithms aim to benefit from rich high-frequency details of an external high-quality reference image such as video frames [13,14] or similar web images [15] during the reconstruction, and many RefSR methods attempt to align and combine information from a low-resolution (LR) input image and a high-resolution (HR) reference image to synthesize a HR image. To this end, most of the studies so far have explored how to find similar features and match the features [16–18] of the LR image and the reference image well. For instance, patch matching [19], deformable convolution [20], and attention [21] techniques have been utilized. The aforementioned methods have succeeded in transferring the high-frequency detail of a reference image. However, these reference-based algorithms show performance degradation when irrelevant high-resolution images are given as references. To this end, we present an online learning technique inspired by zero-shot super-resolution (ZSSR) [22]. In ZSSR, a LR input image

$I^{LR}$ and its downsampled version $I^{LR}\downarrow$ are used for supervision during the inference phase. However, ZSSR has difficulties when dealing with a large scaling factor (e.g., $\times 3$, $\times 4$).

Therefore, in this paper, we propose a method to effectively exploit both LR and HR reference images for online learning to update not only RefSR but also SISR models. Furthermore, using a pre-trained SR model, we create a pseudo-HR image from a LR input image, then use this pair of a pseudo-HR images $\bar{I}^{HR}$ and a downsampled pseudo-HR image $\bar{I}^{HR}\downarrow$ as another datum for the online learning of the SR model. In summary, we perform online learning for both SISR and RefSR models by utilizing three types of supervision, including $I^{LR}$, $I^{Ref}$, and $\bar{I}^{HR}$. We use not only each supervision individually, but also combinations of each supervision for online learning. As a result, the proposed method can benefit from more images during the online adaptation, compared with ZSSR [22].

Our key contributions can be summarized as follows:

- We propose an online learning method for reference-based super-resolution with various data pairs for supervision. To this end, we present three methods for SISR models and four methods for RefSR models;
- Our method is very simple, but it is effective, and can be seamlessly combined with both SISR and RefSR models;
- Our method shows consistent performance improvements without being significantly affected by the degree of similarity between the reference and input images.

## 2. Related Works

In this section, we review deep learning-based SISR and RefSR methods. Then, we introduce recent SR approaches using online adaptation.

Single-image super-resolution (SISR) restores high-frequency details of a LR image using only an input LR image. The traditional SISR approaches [23,24] usually exploit the self-similarity or self-recurrence of the input LR image. With deep learning, Dong et al. [25] introduced a SISR model with just three convolutional layers and outperformed traditional SISR methods with large margins. Kim et al. [3,4] increased the SR performance in terms of PSNR and SSIM by using very deep convolutional layers. Lai et al. [26] suggested LapSRN, which progressively restores high-frequency details with the Laplacian pyramid. Lim et al. [5] does away with unnecessary modules in residual networks such as batch normalization layers, and achieved improved performance. Zhang et al. [7] introduced a channel attention model to take care of the inter-dependency across different channels. The aforementioned methods have significantly increased SR performance in terms of PSNR and SSIM. However, these methods may be blurry or visually unpleasing to human eyes. To enhance visual quality, Johnson et al. [12] proposed perceptual loss that minimizes errors on high-level features. Ledig et al. [11] adopted the GAN framework to generate photo-realistic images. Furthermore, Wang et al. [27] introduced a relativistic adversarial loss based on a residual-in-residual dense block to produce more realistic images. Perception-based methods have succeeded in producing visually good results, but there is a limit to recovering information lost during the downsampling process.

Unlike SISR, reference-based image super-resolution (RefSR) uses an additional HR reference image as an input to restore high-frequency details. Therefore, information lost in the downsampling process can be obtained from the reference image. Zheng et al. [19] proposed RefSR-Net to combine information of both LR and reference images based on patch matching. Specifically, RefSR-Net extracts local patches from both LR and reference images and then searches for correspondences between them. After that, the resulting matches are used to synthesize a HR image. However, the patch-match-based approach has difficulty with handling non-rigid deformations and, thus, suffers from blur or grid artifacts. Using optical flow [28,29] for pixel-wise matching, Zheng et al. [30] presented CrossNet, combining a warping process and image synthesis. CrossNet can effectively handle non-rigid deformations between the input and reference images; however, it is vulnerable to large displacements. With the recent progress of neural-style transfer [31,32], Zhang et al. [33] proposed SRNTT, performing texture transfer from the reference image.

This is particularly robust when an unrelated reference image is paired with an input image. Shim et al. [20] proposed SSEN, which aligns features of input and reference images using non-local blocks and deformable convolutions. Intra-image global similarities extracted from non-local blocks are utilized to estimate relative offset to relevant reference features. After that, deformable convolution operations are used to align reference features to those from the input low-resolution image. It is an end-to-end trainable network that does not require optical flow estimation or explicit patch-match. Yang et al. [21] introduced an attention-based RefSR method called TTSR, and have achieved significant performance improvements. Recently, Jiang et al. [34] presented a knowledge-distillation technique to solve the matching difficulties caused by the scale difference between the reference image and the LR input image. Lu et al. [35] introduced the MASA network for addressing the computational burden problem that may occur in LR image and reference image matching. However, these methods are sensitive to the similarity of the reference image.

Recently, Shocher et al. [22] proposed a zero-shot super-resolution that makes the CNN model flexibly adapt to the test image. In other words, parameters of the CNN model are updated during the test phase, and a CNN model optimized for the input image can be obtained. Furthermore, to reduce the number of updates, meta-learning techniques are applied in [36,37]. In this paper, as the first study to apply online learning to the RefSR problem, we achieved robust RefSR despite the difference in similarity between input and reference images.

## 3. Methods

In this section, we introduce our online learning methods for the RefSR problem using both SISR and RefSR models. We then describe the inference process of our method.

### 3.1. Online Learning

In online learning, the most important point is how to exploit input data given at the test phase. Online learning exploits self-similarity in the image. Using the characteristics of online learning to learn internal features, we proceed with high-resolution reference images with similar characteristics and use them to recover test images to improve performance.

For the RefSR problem, this is more crucial, because two kinds of input data are available: a LR image $I^{LR}$ and the reference image $I^{Ref}$. Therefore, we develop various methods to construct pairs of train-input $X$ and train-target (supervision) $Y$ from multiple images (i.e., $I^{LR}$, $I^{Ref}$). Although our main goal is to solve the RefSR problem in this work, we present methods to construct pairs of data $D$ which can be used to train not only RefSR but also SISR models at the test time.

### 3.1.1. SISR Model

Existing SISR models require data pairs $D_s$ consisting of an input $X$ and supervision $Y$ (i.e., $D_s = \{X, Y\}$) for training. To be specific, we present three strategies, $D_s^{LR}, D_s^{Pse}$, and $D_s^{Ref}$, to construct $D_s$. First, $D_s^{LR}$ consists of a downsampled LR image and an input LR image and is denoted by $D_s^{LR} = \{X : I^{LR} \downarrow, Y : I^{LR}\}$. Note that this is a commonly used configuration to exploit self-similarity in SISR such as ZSSR [22]. Next, $D_s^{Pse}$ is constructed with a pseudo-HR image $\bar{I}^{HR}$ obtained from a pre-trained SR model $\mathbf{P}_\phi(\cdot)$ as follows:

$$\bar{I}^{HR} = \mathbf{P}_\phi(I^{LR}), \tag{1}$$

where $\phi$ is the pre-trained network parameter. Then, we downsample $\bar{I}^{HR}$ to construct a pair of training samples, and the set is defined as $D_s^{Pse} = \{X : \bar{I}^{HR} \downarrow, Y : \bar{I}^{HR}\}$. Finally, we utilize a reference image for $D_s^{Ref}$. Similar to $D_s^{LR}$ and $D_s^{Pse}$, a downsampled image and the original reference images are paired as $D_s^{Ref} = \{X : I^{Ref} \downarrow, Y : I^{Ref}\}$. Using these three

data pairs acquired in the test phase, pre-trained parameters $\theta_s$ of a SISR model $\mathbf{SISR}_{\theta_s}(\cdot)$ are updated by minimizing the following loss function:

$$\mathcal{L}_{\theta_s}(x,y) = \mathbf{E}\big[||\mathbf{SISR}_{\theta_s}(x) - y)||\big], \tag{2}$$

where $x$ and $y$ are extracted patches from $X$ and $Y$ in $D_s$. Note that the aforementioned data pairs can be used individually or in combination.

### 3.1.2. RefSR Model

In order to train RefSR models, a pair of data $D_r$ that comprises an input $X$, a reference $R$, and supervision $Y$ (i.e., $D_r = \{X, R, Y\}$) is required. Thanks to the additional input $R$, it is possible to construct more diverse data pairs than SISR models, and we propose four methods to construct $D_r$: $D_r^{LR}$, $D_r^{Pse}$, $D_r^{Ref1}$, and $D_r^{Ref2}$. First, $D_r^{LR}$ is composed of a downsampled LR image, a reference image, and an input LR image, and denoted by $D_r^{LR} = \left\{ X : I^{LR} \downarrow, R : I^{Ref}, Y : I^{LR} \right\}$ (cf., $D_s^{LR}$). Similarly, we define the second data pair $D_r^{Pse} = \left\{ X : \bar{I}^{HR} \downarrow, R : I^{Ref}, Y : \bar{I}^{HR} \right\}$ by including $I^{Ref}$ to $D_s^{Pse}$ as a reference $R$. In addition, we can utilize a downsampled reference image $I^{Ref} \downarrow$ and the original one $I^{Ref}$ as an input $X$ and supervision $Y$, respectively, and an input LR image as a reference $R$ to make the third data pair $D_r^{Ref1} = \left\{ X : I^{Ref} \downarrow, R : I^{LR}, Y : I^{Ref} \right\}$. Finally, we replace $I^{LR}$ in $D_r^{Ref1}$ with $\bar{I}^{HR}$ to make the last data pair $D_r^{Ref2} = \left\{ X : I^{Ref} \downarrow, R : \bar{I}^{HR}, Y : I^{Ref} \right\}$. Note that $D_r^{Ref1}$ and $D_r^{Ref2}$ are extended from $D_s^{Ref}$ by adding $I^{LR}$ and $\bar{I}^{HR}$ as the reference. With these data pairs, we can update network parameters $\theta_r$ of the pre-trained RefSR model $\mathbf{RefSR}_{\theta_r}(\cdot)$ with the following loss function:

$$\mathcal{L}_{\theta_r}(x,r,y) = \mathbf{E}\big[||\mathbf{RefSR}_{\theta_r}(x,r) - y)||\big], \tag{3}$$

where $x$, $r$ and $y$ are extracted patches from $X$, $R$, and $Y$ in $D_r$. Similar to SISR models, data pairs can be used individually or combined for online learning. The data pairs for online learning of SISR and RefSR models are summarized in Table 1.

**Table 1.** Online learning data pairs for SISR and RefSR models.

| Model | SISR | | | RefSR | | | |
|---|---|---|---|---|---|---|---|
| Data Pair | $D_s^{LR}$ | $D_s^{Pse}$ | $D_s^{Ref}$ | $D_r^{LR}$ | $D_r^{Pse}$ | $D_r^{Ref1}$ | $D_r^{Ref2}$ |
| X | $I^{LR} \downarrow$ | $\bar{I}^{HR} \downarrow$ | $I^{Ref} \downarrow$ | $I^{LR} \downarrow$ | $\bar{I}^{HR} \downarrow$ | $I^{Ref} \downarrow$ | $I^{Ref} \downarrow$ |
| R | - | - | - | $I^{Ref}$ | $I^{Ref}$ | $I^{LR}$ | $\bar{I}^{HR}$ |
| Y | $I^{LR}$ | $\bar{I}^{HR}$ | $I^{Ref}$ | $I^{LR}$ | $\bar{I}^{HR}$ | $I^{Ref}$ | $I^{Ref}$ |

### 3.2. Inference

With the updated parameters of SISR or RefSR models at the test stage, we estimate the final super-resolved output image as follows:

$$\bar{O}_s = \mathbf{SISR}_{\theta_s}(I^{LR}), \quad \bar{O}_r = \mathbf{RefSR}_{\theta_r}(I^{LR}, I^{Ref}). \tag{4}$$

Notably, unlike RefSR, SISR models are updated using the reference image in the online learning phase, but the reference image is not used for the final inference.

## 4. Experiments

In this section, we describe implementation details and demonstrate both quantitative and qualitative comparisons with existing methods. We also provide various empirical analyses, including experiments according to the similarity between the reference image and the input LR image and experiments using non-bicubic degradation LR images.

### 4.1. Implementation Details

For both SISR and RefSR models, we used the CUFED dataset [33], which consists of 11,871 pairs of input and reference images to pre-train the models for $\times 4$ upscaling. As baseline SISR models, we have adopted light-weight versions of SimpleNet [22], RCAN [7], and EDSR [5] for fast execution time. Specifically, the number of residual blocks is reduced from 20 to 6 for the RCAN, and from 32 to 16 for EDSR with 64 feature dimensions. Each model is trained for 100 epochs with 32 batch sizes. We use all training data, including both HR and reference images. For RefSR models, SSEN [20] and TTSR [21] are adopted. SSEN is trained for 200 epochs with a batch size of 32, and TTSR is trained for 200 epochs with a batch size of 9. In the online learning phase, the CUFED5 dataset [33] is used. It consists of 126 groups of images, and each group contains a HR image and 5 reference images with different levels of similarity. Images are augmented with random crop ($128 \times 128$), rotation, and flip. The initial learning rate is set to $1 \times 10^{-4}$ for ADAM, and we multiply by 0.1 when the loss values stop decreasing [22]. Our method is implemented using PyTorch on Ubuntu 16.04 with a single RTX 2080 GPU.

### 4.2. Experimental Results

For all experiments, we have trained SISR and RefSR models by following their original configurations to obtain the baseline models. After that, the proposed online learning is applied to verify the effectiveness of our algorithm. Note that our method does not introduce any additional modules to the baseline models. All the models are evaluated on the CUFED5 test set. We evaluate in terms of PSNR, SSIM, and LPIPS [12], and the LPIPS value is measured with the VGG model.

Table 2 shows SISR online learning results over the baseline models. Online learning with only $D_s^{LR}$ degrades performance in all models because the size of $D_s^{LR}$ is too small ($30 \times 20$) to exploit abundant information. In contrast, $D_s^{Pse}$ contains plenty of HR details useful for the inference and, thus, performance is consistently improved with $D_s^{Pse}$ as in [38]. Notably, we see further improvement by combining $D_s^{LR}$ with $D_s^{Pse}$. Different from the results using $D_s^{LR}$ only, $D_s^{LR} + D_s^{Pse}$ can effectively benefit from $D_s^{LR}$ for self-similarity while keeping knowledge of HR information from $D_s^{Pse}$.

**Table 2.** SISR online learning results on SISR models.

| Model | Method | PSNR | SSIM | LPIPS |
|-------|--------|------|------|-------|
| SRCNN [9] | Pre-trained | 25.475 | 0.737 | 0.3369 |
| | $D_s^{LR}$ | 25.379 | 0.732 | 0.3388 |
| | $D_s^{Pse}$ | 25.563 | 0.741 | 0.3273 |
| | $D_s^{LR} + D_s^{Pse}$ | 25.559 | 0.741 | 0.3275 |
| VDSR [3] | Pre-trained | 25.660 | 0.746 | 0.3332 |
| | $D_s^{LR}$ | 25.500 | 0.740 | 0.3229 |
| | $D_s^{Pse}$ | 25.709 | 0.748 | 0.3256 |
| | $D_s^{LR} + D_s^{Pse}$ | 25.734 | 0.749 | 0.3245 |
| SimpleNet [22] | Pre-trained | 25.800 | 0.753 | 0.3267 |
| | $D_s^{LR}$ | 25.727 | 0.750 | 0.3128 |
| | $D_s^{Pse}$ | 25.941 | 0.757 | 0.3152 |
| | $D_s^{LR} + D_s^{Pse}$ | 25.958 | 0.757 | 0.3136 |
| EDSR [5] | Pre-trained | 26.198 | 0.771 | 0.2955 |
| | $D_s^{LR}$ | 26.132 | 0.765 | 0.2897 |
| | $D_s^{Pse}$ | 26.422 | 0.774 | 0.2956 |
| | $D_s^{LR} + D_s^{Pse}$ | 26.440 | 0.775 | 0.2932 |
| RCAN [7] | Pre-trained | 26.243 | 0.774 | 0.2906 |
| | $D_s^{LR}$ | 26.147 | 0.767 | 0.2883 |
| | $D_s^{Pse}$ | 26.500 | 0.777 | 0.2912 |
| | $D_s^{LR} + D_s^{Pse}$ | 26.512 | 0.778 | 0.2892 |

RefSR online learning results on the SISR baseline models are shown in Table 3. In RefSR online learning, baseline models always show performance improvements with $D_s^{Ref}$ because it contains real high-frequency details not available in $D_s^{Pse}$. We achieve the best results by using both $D_s^{LR}$ and $D_s^{Ref}$, rather than using either $D_s^{LR}$ or $D_s^{Ref}$, respectively. Similar results are observed with RefSR online learning on the RefSR models where the best performance is mostly achieved with $D_r^{Pse} + D_r^{Ref2}$, as shown in Table 4. Figure 1 shows qualitative comparisons between existing methods and ours. Note that Figure 1f,k show superior performance over their baseline counterparts Figure 1e,j.

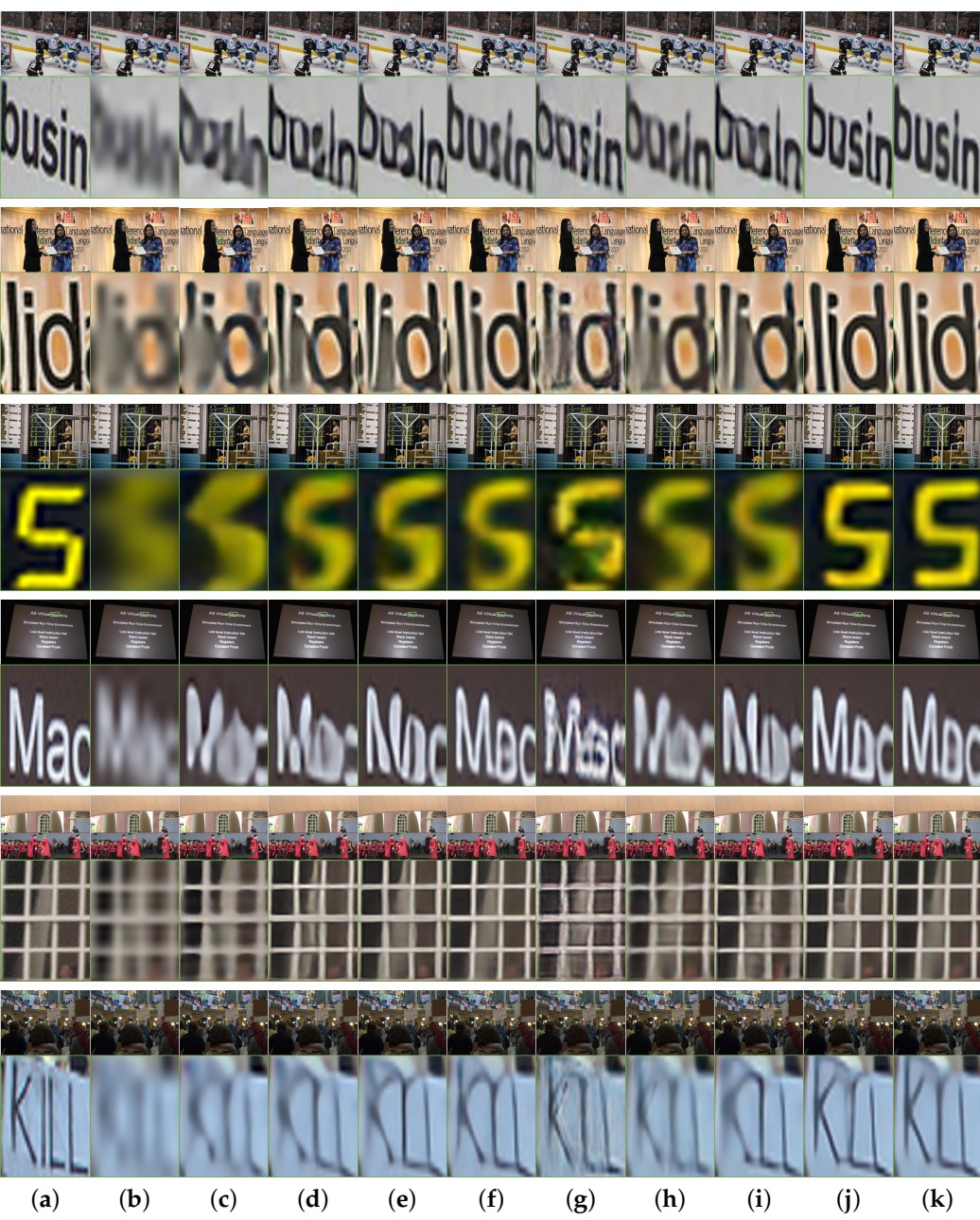

|   |   |   |   |   |   |   |   |   |   |   |
|---|---|---|---|---|---|---|---|---|---|---|
| (a) | (b) | (c) | (d) | (e) | (f) | (g) | (h) | (i) | (j) | (k) |

**Figure 1.** Qualitative comparison of RefSR methods on the CUFED5 datasets. (**a**) GT images. (**b–k**) Results of Bicubic, SimpleNet [22], EDSR [5], RCAN [7], Ours+RCAN [7], SRNTT [33], SRNTT-$\ell_2$ [33], SSEN [20], TTSR-*rec* [21], and Ours+TTSR-*rec* [21], respectively.

**Table 3.** RefSR online learning results on SISR models.

| Model | Method | Similarity | | | | | | | | | | | | | | |
|---|---|---|---|---|---|---|---|---|---|---|---|---|---|---|---|---|
| | | XL | | | L | | | M | | | H | | | XH | | |
| | | PSNR | SSIM | LPIPS | PSNR | SSIM | LPIPS | PSNR | SSIM | LPIPS | PSNR | SSIM | LPIPS | PSNR | SSIM | LPIPS |
| Ours +<br>SimpleNet [22] | $D_s^{Ref}$ | 25.888 | 0.755 | 0.3159 | 25.932 | 0.756 | 0.3153 | 25.925 | 0.757 | 0.3148 | 25.990 | 0.758 | 0.3140 | 26.046 | 0.760 | 0.3127 |
| | $D_s^{LR} + D_s^{Ref}$ | 25.894 | 0.755 | 0.3134 | 25.960 | 0.757 | 0.3119 | 25.950 | 0.758 | 0.3115 | 26.003 | 0.758 | 0.3107 | 26.058 | 0.761 | 0.3093 |
| | $D_s^{Pse} + D_s^{Ref}$ | 25.936 | 0.757 | 0.3147 | 25.963 | 0.758 | 0.3145 | 25.979 | 0.758 | 0.3143 | 25.985 | 0.758 | 0.3140 | 26.018 | 0.759 | 0.3134 |
| | $D_s^{LR} + D_s^{Pse} + D_s^{Ref}$ | 25.973 | 0.758 | 0.3131 | 25.991 | 0.758 | 0.3130 | 25.997 | 0.759 | 0.3130 | 26.010 | 0.759 | 0.3129 | 26.049 | 0.760 | 0.3122 |
| Ours +<br>EDSR [5] | $D_s^{Ref}$ | 26.354 | 0.772 | 0.2959 | 26.418 | 0.773 | 0.2937 | 26.417 | 0.774 | 0.2932 | 26.512 | 0.776 | 0.2922 | 26.645 | 0.780 | 0.2888 |
| | $D_s^{LR} + D_s^{Ref}$ | 26.385 | 0.773 | 0.2889 | 26.438 | 0.775 | 0.2875 | 26.444 | 0.775 | 0.2875 | 26.553 | 0.777 | 0.2861 | 26.699 | 0.782 | 0.2833 |
| | $D_s^{Pse} + D_s^{Ref}$ | 26.452 | 0.775 | 0.2949 | 26.467 | 0.775 | 0.2944 | 26.500 | 0.776 | 0.2935 | 26.497 | 0.776 | 0.2938 | 26.559 | 0.778 | 0.2926 |
| | $D_s^{LR} + D_s^{Pse} + D_s^{Ref}$ | 26.462 | 0.775 | 0.2925 | 26.484 | 0.776 | 0.2922 | 26.508 | 0.776 | 0.2916 | 26.522 | 0.776 | 0.2917 | 26.577 | 0.778 | 0.2902 |
| Ours +<br>RCAN [7] | $D_s^{Ref}$ | 26.402 | 0.773 | 0.2919 | 26.465 | 0.775 | 0.2906 | 26.465 | 0.775 | 0.2900 | 26.581 | 0.778 | 0.2886 | 26.703 | 0.782 | 0.2856 |
| | $D_s^{LR} + D_s^{Ref}$ | 26.418 | 0.774 | 0.2862 | 26.499 | 0.777 | 0.2853 | 26.505 | 0.777 | 0.2845 | 26.635 | 0.780 | 0.2828 | 26.810 | 0.785 | 0.2796 |
| | $D_s^{Pse} + D_s^{Ref}$ | 26.511 | 0.777 | 0.2908 | 26.547 | 0.778 | 0.2901 | 26.567 | 0.778 | 0.2901 | 26.589 | 0.779 | 0.2895 | 26.634 | 0.781 | 0.2887 |
| | $D_s^{LR} + D_s^{Pse} + D_s^{Ref}$ | 26.543 | 0.778 | 0.2890 | 26.562 | 0.779 | 0.2885 | 26.574 | 0.779 | 0.2884 | 26.607 | 0.780 | 0.2877 | 26.681 | 0.782 | 0.2862 |

**Table 4.** RefSR online learning results on RefSR models.

| Model | Method | Similarity | | | | | | | | | | | | | | |
|---|---|---|---|---|---|---|---|---|---|---|---|---|---|---|---|
| | | XL | | | L | | | M | | | H | | | XH | | |
| | | PSNR | SSIM | LPIPS | PSNR | SSIM | LPIPS | PSNR | SSIM | LPIPS | PSNR | SSIM | LPIPS | PSNR | SSIM | LPIPS |
| SRNTT [33] | Pre-trained | 25.14 | 0.729 | 0.2476 | 25.07 | 0.720 | 0.2410 | 25.06 | 0.728 | 0.2354 | 25.13 | 0.734 | 0.2294 | 25.17 | 0.734 | 0.2099 |
| SRNTT-$\ell_2$ [33] | Pre-trained | 25.87 | 0.757 | 0.2949 | 25.88 | 0.758 | 0.2916 | 25.90 | 0.758 | 0.2893 | 25.97 | 0.760 | 0.2856 | 26.06 | 0.765 | 0.2758 |
| SSEN [20] | Pre-trained | 26.156 | 0.768 | 0.2979 | 26.151 | 0.768 | 0.2980 | 26.149 | 0.768 | 0.2979 | 26.154 | 0.768 | 0.2977 | 26.152 | 0.769 | 0.2976 |
| Ours + SSEN [20] | $D_r^{LR}$ | 26.109 | 0.764 | 0.2879 | 26.107 | 0.764 | 0.2881 | 26.116 | 0.764 | 0.2889 | 26.108 | 0.764 | 0.2883 | 26.112 | 0.764 | 0.2884 |
| | $D_r^{Pse}$ | 26.434 | 0.774 | 0.2951 | 26.459 | 0.775 | 0.2946 | 26.480 | 0.775 | 0.2944 | 26.480 | 0.775 | 0.2940 | 26.527 | 0.777 | 0.2930 |
| | $D_r^{Ref1}$ | 26.226 | 0.767 | 0.2931 | 26.206 | 0.768 | 0.2925 | 26.241 | 0.768 | 0.2921 | 26.284 | 0.769 | 0.2903 | 26.276 | 0.770 | 0.2895 |
| | $D_r^{Ref2}$ | 26.343 | 0.771 | 0.2946 | 26.383 | 0.772 | 0.2936 | 26.475 | 0.774 | 0.2920 | 26.509 | 0.775 | 0.2911 | 26.675 | 0.780 | 0.2874 |
| | $D_r^{LR} + D_r^{Ref1}$ | 26.205 | 0.767 | 0.2852 | 26.206 | 0.767 | 0.2856 | 26.221 | 0.767 | 0.2854 | 26.261 | 0.768 | 0.2843 | 26.257 | 0.769 | 0.2946 |
| | $D_r^{Pse} + D_r^{Ref2}$ | 26.392 | 0.773 | 0.2955 | 26.460 | 0.774 | 0.2942 | 26.475 | 0.774 | 0.2946 | 26.505 | 0.775 | 0.2935 | 26.568 | 0.777 | 0.2924 |
| TTSR-*rec* [21] | Pre-trained | 26.586 | 0.783 | 0.2825 | 26.623 | 0.785 | 0.2800 | 26.685 | 0.787 | 0.2782 | 26.787 | 0.789 | 0.2759 | 27.039 | 0.799 | 0.2653 |
| Ours + TTSR-*rec* [21] | $D_r^{LR}$ | 26.407 | 0.775 | 0.2711 | 26.455 | 0.776 | 0.2689 | 26.502 | 0.778 | 0.2675 | 26.579 | 0.780 | 0.2643 | 26.812 | 0.788 | 0.2545 |
| | $D_r^{Pse}$ | 26.822 | 0.786 | 0.2815 | 26.866 | 0.788 | 0.2792 | 26.937 | 0.790 | 0.2782 | 27.027 | 0.791 | 0.2760 | 27.337 | 0.801 | 0.2663 |
| | $D_r^{Ref1}$ | 26.540 | 0.778 | 0.2791 | 26.563 | 0.781 | 0.2757 | 26.622 | 0.782 | 0.2750 | 26.769 | 0.785 | 0.2712 | 26.986 | 0.794 | 0.2614 |
| | $D_r^{Ref2}$ | 26.658 | 0.782 | 0.2818 | 26.717 | 0.785 | 0.2788 | 26.836 | 0.787 | 0.2757 | 26.959 | 0.790 | 0.2730 | 27.383 | 0.802 | 0.2578 |
| | $D_r^{LR} + D_r^{Ref1}$ | 26.497 | 0.777 | 0.2696 | 26.522 | 0.779 | 0.2668 | 26.592 | 0.780 | 0.2660 | 26.698 | 0.782 | 0.2635 | 26.900 | 0.790 | 0.2529 |
| | $D_r^{Pse} + D_r^{Ref2}$ | 26.845 | 0.786 | 0.2816 | 26.877 | 0.788 | 0.2796 | 26.980 | 0.790 | 0.2780 | 27.056 | 0.792 | 0.2760 | 27.400 | 0.801 | 0.2663 |

*4.3. Empirical Analyses*

- **Reference Similarity**

We first analyze the effect of similarity of reference images on online learning. The CUFED5 dataset [33] provides five similarity levels, from the lowest (i.e., XL) to the highest (i.e., XH), depending on the content similarity between the reference and LR images. For both SISR and RefSR models, performance improvement is proportional to the level of similarity, and the best performance is obtained with the reference with the highest similarity XH. This result is expected, because XH reference images contain a large amount of real high-frequency details closely related to the lost details of LR images. Therefore, online learning with XH reference images can train baseline models with strong and relevant HR guidance. On the contrary, the amount of relevant information is reduced with decreasing similarity of the reference images; therefore, performance improvement also decreases, as shown in Tables 3 and 4.

- **Pseudo HR vs. LR for Supervision**

For RefSR online learning for SISR models in Table 3, we can compare two results by $D_s^{LR} + D_s^{Ref}$ and $D_s^{Pse} + D_s^{Ref}$. With high similarity levels (i.e., XH and H), $D_s^{LR} + D_s^{Ref}$ shows better performance while $D_s^{Pse} + D_s^{Ref}$ works better for low similarity levels (i.e., M, L, and XL). For XH and H reference images, baseline networks can exploit highly relevant information from them, as we inspected. However, the role of $D_s^{Ref}$ is weakened with irrelevant reference images, while that of the combined data, $D_s^{LR}$ or $D_s^{Pse}$, is relatively emphasized. Therefore, the performance improvement from $D_s^{Ref}$ combined with $D_s^{Pse}$ is superior, thanks to the knowledge from a pre-trained model (i.e., $D_s^{Pse}$), compared to the self-supervision (i.e., $D_s^{LR}$). Meanwhile, for RefSR online learning for RefSR models, fine-tuning with $\bar{I}^{HR}$ achieves better overall performance than fine-tuning with $I^{LR}$. In other words, for all similarity levels, $D_r^{Pse} + D_r^{Ref2}$ shows better performance than $D_r^{LR} + D_r^{Ref1}$, as reported in Table 4. The reason for this is that $\bar{I}^{HR}$ has a relatively similar resolution to $I^{Ref}$ than $I^{LR}$; thus, it is much easier to align and combine with the information in the reference image.

- **Non-Bicubic Degradation**

We further validate our algorithm with a LR image with non-bicubic degradation for both RCAN (SISR) and TTSR (RefSR) models. For non-bicubic $\times 4$ degradation, we have utilized isotropic ($g_w$) and anisotropic ($g_{ani}$) Gaussian kernels of width $w$ with direct ($g^d$) and bicubic ($g^b$) subsampling methods presented in MZSR [36]. Table 5 shows that RCAN and TTSR pre-trained with bicubic degradation produce inferior SR results because they cannot handle non-bicubic degradation. However, RCAN and TTSR models can achieve substantial performance gains with the proposed online learning if the non-bicubic degradation model is given during the online learning (i.e., Non-blind). Moreover, RCAN and TTSR can be improved during the online learning in a blind manner that is conducted using each input obtained by downsampling with a random kernel [36]. Figure 2 shows qualitative non-bicubic comparisons between existing methods and ours. Therefore, we conclude that the proposed method can handle any type of degradation (i.e., bicubic and non-bicubic), regardless of the awareness of the degradation kernel (i.e., blind and non-blind).

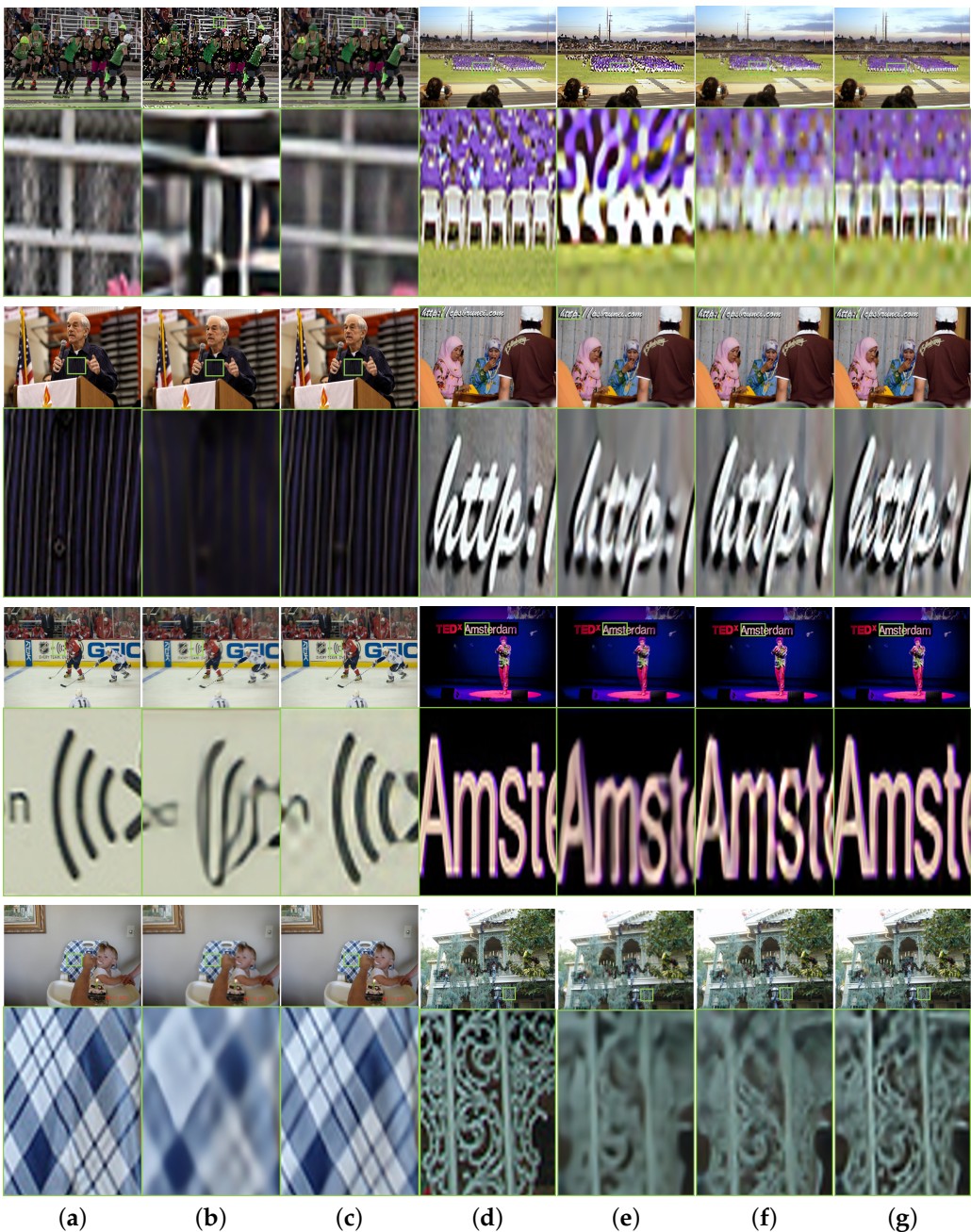

**Figure 2.** Qualitative comparison of RefSR methods on the non-bicubic CUFED5 datasets. From the top, each kernel ($g_{0.2}^d$, $g_{2.0}^d$, $g_{aig}^d$, $g_{1.3}^b$) was used. (**a**) GT images. (**b**) RCAN [7]. (**c**) Ours + RCAN [7]. (**d**) GT images. (**e**) TTSR [21]. (**f**) Ours + TTSR [21] using $D_r^{Ref1}$. (**g**) Ours + TTSR [21] using $D_r^{Ref2}$.

**Table 5.** Online learning results with non-bicubic degradation.

| Model | Kernel | Blind | Method | PSNR | SSIM | LPIPS |
|---|---|---|---|---|---|---|
| Ours+ EDSR [5] | $g^d_{0.2}$ | - | Pre-trained | 18.754 | 0.534 | 0.4068 |
| | | Non-blind | $D_s^{LR}+D_s^{Ref}$ | 24.335 | 0.726 | 0.3035 |
| | $g^d_{2.0}$ | - | Pre-trained | 21.387 | 0.606 | 0.3771 |
| | | Non-blind | $D_s^{LR}+D_s^{Ref}$ | 26.263 | 0.772 | 0.2665 |
| | $g^d_{ani}$ | - | Pre-trained | 21.364 | 0.593 | 0.3639 |
| | | Non-blind | $D_s^{LR}+D_s^{Ref}$ | 26.164 | 0.765 | 0.2764 |
| | $g^b_{1.3}$ | - | Pre-trained | 25.595 | 0.741 | 0.3288 |
| | | Non-blind | $D_s^{LR}+D_s^{Ref}$ | 26.655 | 0.780 | 0.2663 |
| | - | - | Pre-trained | 21.896 | 0.608 | 0.3892 |
| | | Blind | $D_s^{LR}+D_s^{Ref}$ | 24.354 | 0.697 | 0.3459 |
| Ours+ RCAN [7] | $g^d_{0.2}$ | - | Pre-trained | 17.938 | 0.497 | 0.4111 |
| | | Non-blind | $D_s^{LR}+D_s^{Ref}$ | 24.532 | 0.737 | 0.2910 |
| | $g^d_{2.0}$ | - | Pre-trained | 21.131 | 0.597 | 0.3335 |
| | | Non-blind | $D_s^{LR}+D_s^{Ref}$ | 26.545 | 0.783 | 0.2586 |
| | $g^d_{ani}$ | - | Pre-trained | 21.198 | 0.587 | 0.3609 |
| | | Non-blind | $D_s^{LR}+D_s^{Ref}$ | 26.414 | 0.775 | 0.2679 |
| | $g^b_{1.3}$ | - | Pre-trained | 25.484 | 0.738 | 0.3314 |
| | | Non-blind | $D_s^{LR}+D_s^{Ref}$ | 26.909 | 0.790 | 0.2597 |
| | - | - | Pre-trained | 21.798 | 0.606 | 0.3914 |
| | | Blind | $D_s^{LR}+D_s^{Ref}$ | 24.277 | 0.692 | 0.3480 |

| Model | Kernel | Blind | Method | PSNR | SSIM | LPIPS |
|---|---|---|---|---|---|---|
| Ours+ SSEN [20] | $g^d_{0.2}$ | - | Pre-trained | 18.538 | 0.521 | 0.4142 |
| | | Non-blind | $D_r^{Ref1}$ | 23.565 | 0.694 | 0.3431 |
| | | | $D_r^{Ref2}$ | 24.155 | 0.720 | 0.3239 |
| | $g^d_{2.0}$ | - | Pre-trained | 20.706 | 0.586 | 0.3541 |
| | | Non-blind | $D_r^{Ref1}$ | 25.436 | 0.741 | 0.2891 |
| | | | $D_r^{Ref2}$ | 26.105 | 0.765 | 0.2838 |
| | $g^d_{ani}$ | - | Pre-trained | 21.269 | 0.590 | 0.3633 |
| | | Non-blind | $D_r^{Ref1}$ | 25.213 | 0.728 | 0.3062 |
| | | | $D_r^{Ref2}$ | 25.882 | 0.753 | 0.2953 |
| | $g^b_{1.3}$ | - | Pre-trained | 25.522 | 0.740 | 0.3273 |
| | | Non-blind | $D_r^{Ref1}$ | 26.010 | 0.758 | 0.2773 |
| | | | $D_r^{Ref2}$ | 26.569 | 0.778 | 0.2789 |
| | - | - | Pre-trained | 21.836 | 0.606 | 0.3881 |
| | | Blind | $D_r^{Ref1}$ | 23.953 | 0.676 | 0.3654 |
| | | | $D_r^{Ref2}$ | 24.201 | 0.685 | 0.3608 |
| Ours+ TTSR-rec [21] | $g^d_{0.2}$ | - | Pre-trained | 18.415 | 0.524 | 0.4039 |
| | | Non-blind | $D_r^{Ref1}$ | 23.489 | 0.688 | 0.3423 |
| | | | $D_r^{Ref2}$ | 24.168 | 0.717 | 0.3232 |
| | $g^d_{2.0}$ | - | Pre-trained | 21.211 | 0.609 | 0.3127 |
| | | Non-blind | $D_r^{Ref1}$ | 25.911 | 0.760 | 0.2647 |
| | | | $D_r^{Ref2}$ | 26.561 | 0.784 | 0.2624 |
| | $g^d_{ani}$ | - | Pre-trained | 21.199 | 0.596 | 0.3367 |
| | | Non-blind | $D_r^{Ref1}$ | 25.512 | 0.741 | 0.2841 |
| | | | $D_r^{Ref2}$ | 26.199 | 0.768 | 0.2754 |
| | $g^b_{1.3}$ | - | Pre-trained | 26.147 | 0.767 | 0.2912 |
| | | Non-blind | $D_r^{Ref1}$ | 26.599 | 0.781 | 0.2471 |
| | | | $D_r^{Ref2}$ | 26.989 | 0.796 | 0.2471 |
| | - | - | Pre-trained | 21.820 | 0.615 | 0.3603 |
| | | Blind | $D_r^{Ref1}$ | 23.928 | 0.672 | 0.3535 |
| | | | $D_r^{Ref2}$ | 24.010 | 0.684 | 0.3461 |

## 5. Conclusions

We have proposed an online learning algorithm for RefSR to exploit various types of data for network adaptation in the test stage. The proposed method has brought significant performance improvements to both SISR and RefSR models without introducing any additional network parameters. Specifically, various types of data pairs are proposed using input LR, pseudo-HR, and reference HR images, and the role of each data pair is verified with different similarity levels of the reference images. Extensive experimental results demonstrate the validity, efficiency, and versatility of the proposed algorithm.

**Author Contributions:** Conceptualization, D.C.; formal analysis, J.P.; investigation, T.-H.K. and D.C.; data curation, B.C.; writing—original draft preparation, D.C.; writing—review and editing, T.-H.K and D.C.; supervision, D.C. and J.P.; funding acquisition, D.C. All authors have read and agreed to the published version of the manuscript.

**Funding:** This work was supported by BK21 FOUR Program by Chungnam National University Research Grant, 2021–2022.

**Conflicts of Interest:** The authors declare no conflict of interest.

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
