# Peer review of "Online Learning for Reference-Based Super-Resolution"

_electronics, doi:10.3390/electronics11071064_

Round 1

Reviewer 1 Report

1、There are many formatting problems in the article, such as the misplacement of the title row of the table, I hope the author can check it carefully.

2、The experimental part of the article is not perfect, I hope the author can add several comparison algorithms

3、There are some problems with the reference format, hope the author can double check.

4、There are some grammar problems in this article, I hope the author can check it carefully.

5、The author's references are somewhat lacking, I hope the author can introduce three or more related papers in the references, such as:

A novel point-matching algorithm based on fast sample consensus for image registration.

Commonality Autoencoder: Learning Common Features for Change Detection From Heterogeneous Images.

A Two-Step Method for Remote Sensing Images Registration Based on Local and Global Constraints.

Reviewer 2 Report

Authors suggested online learning for superresolution. The idea is reasonable. Authors have presented experiments. This work may be accepted for publication subject to following:

  1. Detail description of methods should be presented. It should clearly point out technical contributions on the online learning front. Algorithm(s) should also be added for better representation.
  2. More results both visual and quantitative would make the manuscript more valuable. In particular, results should highlight how online learning impacted SR processes.
  3. Kindly get it corrected for usage of English language.

Round 2

Reviewer 2 Report

no comment